# Localization of Catecholaminergic Neurofibers in Pregnant Cervix as a Possible Myometrial Pacemaker

**DOI:** 10.3390/ijms25115630

**Published:** 2024-05-22

**Authors:** Antonio Malvasi, Giorgio Maria Baldini, Ettore Cicinelli, Edoardo Di Naro, Domenico Baldini, Alessandro Favilli, Paola Tiziana Quellari, Paola Sabbatini, Bernard Fioretti, Lorenzo E. Malgieri, Gianluca Raffaello Damiani, Miriam Dellino, Giuseppe Trojano, Andrea Tinelli

**Affiliations:** 11st Unit of Gynecology and Obstetrics, Department of Interdisciplinary Medicine (DIM), University of Bari (BA), 70124 Bari, Italy; antoniomalvasi@gmail.com (A.M.); gbaldini97@gmail.com (G.M.B.); ettore.cicinelli@uniba.it (E.C.); edoardo.dinaro@uniba.it (E.D.N.); damiani14@alice.it (G.R.D.); miriamdellino@hotmail.it (M.D.); 2MOMO’ FertiLIFE, IVF Clinic, 76011 Bisceglie, Italy; dbaldini@libero.it; 3Department of Medicine and Surgery, Perugia Medical School, University of Perugia, Piazza Lucio Severi 1, 06132 Perugia, Italy; alessandro.favilli@unipg.it (A.F.); paolatiziana.quellari@studenti.unipg.it (P.T.Q.); 4Department of Chemistry, Biology and Biotechnologies, University of Perugia, Via dell’Elce di Sotto 8, 06132 Perugia, Italy; paola.sabbatini@unipg.it (P.S.); bernard.fioretti@unipg.it (B.F.); 5ASST Grande Ospedale Metropolitano Niguarda, 20162 Milano, Italy; 6Chief Innovation Officer in CLE, 70124 Bari, Italy; lorenzo@malgieri.org; 7Department of Maternal and Child, Madonna delle Grazie Hospital ASM, 75100 Matera, Italy; giuseppe.trojano@asmbasilicata.it; 8Department of Gynaecology and Obstetrics, CERICSAL (CEntro di RIcerca Clinico SALentino), “Veris delli Ponti Hospital”, 73020 Lecce, Italy

**Keywords:** pregnancy, uterine pacemaker, catecholamine, neurotransmitters, cervix, lower uterine segment (LUS), labor, delivery, cesarean section

## Abstract

In eutocic labor, the autonomic nervous system is dominated by the parasympathetic system, which ensures optimal blood flow to the uterus and placenta. This study is focused on the detection of the quantitative presence of catecholamine (C) neurofibers in the internal uterine orifice (IUO) and in the lower uterine segment (LUS) of the pregnant uterus, which could play a role in labor and delivery. A total of 102 women were enrolled before their submission to a scheduled cesarean section (CS); patients showed a singleton fetus in a cephalic presentation outside labor. During CS, surgeons sampled two serial consecutive full-thickness sections 5 mm in depth (including the myometrial layer) on the LUS and two randomly selected samples of 5 mm depth from the IUO of the cervix. All histological samples were studied to quantify the distribution of A nerve fibers. The authors demonstrated a significant and notably higher concentration of A fibers in the IUO (46 ± 4.8) than in the LUS (21 ± 2.6), showing that the pregnant cervix has a greater concentration of A neurofibers than the at-term LUS. Pregnant women’s mechanosensitive pacemakers can operate normally when the body is in a physiological state, which permits normal uterine contractions and eutocic delivery. The increased frequency of C neurofibers in the cervix may influence the smooth muscle cell bundles’ activation, which could cause an aberrant mechano-sensitive pacemaker activation–deactivation cycle. Stressful circumstances (anxiety, tension, fetal head position) cause the sympathetic nervous system to become more active, working through these nerve fibers in the gravid cervix. They might interfere with the mechano-sensitive pacemakers, slowing down the uterine contractions and cervix ripening, which could result in dystocic labor.

## 1. Introduction

The anatomy and innervation of the human pregnant uterine cervix have received few descriptions in the literature. A large number of neurofibers were detected in the cervix of the unpregnant uterus, with a high adrenergic-type concentration [1]. However, the localization and function of all these neurofibers in pregnancy and during labor needs to be further investigated.

Toward the term of pregnancy, the cervix shows an anatomical modification forming the lower uterine segment (LUS) following anatomical, structural, and plastic changes in the uterine isthmus.

The uterine isthmus is a small anatomical region located in the upper portion of the cervical canal, 5 mm in the non-pregnant uterus. Some authors describe the isthmus as the lower neck region of the uterus, continuing downwards into the cervix [2]. The isthmus is delimited by two internal uterine narrowings: the upper one is called the anatomical orifice of Aschoff and the lower one is called the histological internal uterine orifice (Figure 1) [3].

Anatomic Internal Uterine Orifice (AIUO)—Histologic Internal Uterine Orifice (HIUO).

In pregnancy, the isthmus participates in the formation of the LUS, an open cap in the top-shaped anatomical region consisting of the thin walls between the uterine body and the cervix. It only exists and develops during pregnancy, with a maximum size during labor and childbirth, and then disappears after placental delivery. The LUS can reach an extreme size in dystocic labor [4,5], to the point of even having a pathological transformation, visible during dystocic labor [6].

The LUS has been morphologically evaluated in post-CS uteri by transuterine ultrasound (TUS) and, anatomically, by histology [7]. The nerve bundles involved in pregnancy and the activation of labor are of the orthosympathetic and parasympathetic types. The sympathetic supply comes from the superior hypogastric plexus and the ovarian plexus. Parasympathetic innervation comes from the second, third, and fourth sacral nerves (pudendal nerve). The fibers coming from the hypogastric nerve and the pudendal nerve form the inferior hypogastric plexus (formerly Frankenhauser’s ganglion), a rhomboid nerve plate located laterally, on both sides, in the paracervical area, and is responsible for sensory and motor innervation [8].

The sympathetic pathway is the visceral–sensory pathway and is responsible for the motility of the uterine body. The parasympathetic pathway is the visceral–motor pathway responsible for the closure of the internal uterine orifice [9]. The cervix is more innervated than the uterine body, and its innervation remains constant during pregnancy [10]. During pregnancy, the uterus remains quiescent thanks to the greater action of catecholamine on the β2 adrenergic receptors of the myometrium. At the time of labor, it is hypothesized that the reduction in progesterone concentration increases the catabolism of catecholamine and reduces the density of β2 adrenergic receptors, while estradiol increases the production of norepinephrine at the level of the sympathetic terminals, causing an α1 adrenergic predominance that favors the contractile activity of the myometrium. Furthermore, the stimulation of sensory fibers, particularly represented at the cervical level, secondary to the stretching and distension of the lower uterine segment, causes a neuroendocrine reflex, called the Ferguson reflex, which indicates the production of oxytocin from the neurohypophysis. This mechanism contributes to the progressive increase in myometrial contractions throughout labor and favors the dilation of the uterine cervix [10].

Furthermore, even though the uterine body provides the environment for the fetus to grow during pregnancy, the cervix must remain closed to prevent early delivery; it is only during term pregnancy that the cervix softens and undergoes what is known as “ripening”, which ultimately leads to dilatation [11]. The ripening of the uterine cervix is the primary trigger for the activation of physiological labor, acting as a pacemaker. The structure and operation of a pacemaker—as a generator of electrical impulses stimulating uterine contraction—in the cervix during pregnancy and the LUS remain unclear in the literature. The existence, number, localization, and physiological mechanism of these visceral pacemakers have been studied and have been the objects of debate in the scientific literature for several years since some authors believe in the existence of single uterine pacemakers [12,13]. Young et al. in a review reported that a pregnant uterus has multiple widespread mechanically sensitive and functional pacemakers; they suggested that uterine contraction coordination during labor occurs via intrauterine pressure activation of many mechanosensitive electrogenic pacemakers, successively stimulating the active mechanic pacemakers and, thus, uterine contraction [14]. In uterine samples from women undergoing emergency CS, Malvasi et al. showed a decrease in neuropeptides, indicating that a prolonged fetal head station in LUS can cause tissue denervation following tissue overdistention and inflammation [15]. Our study is focused on a pregnant internal uterine orifice (IUO) and LUS, which could play a physiological active role in favoring labor and delivery. In particular, we studied the localization of catecholaminergic neurofibers in the uterine cervix, which may act as pacemakers of the myometrium in the activation of labor.

## 2. Results

A total of 102 women were enrolled and all of them completed the study. The mean age of the patients was 31.4 ± 4.4, the mean BMI was 24.7 ± 7.3 kg/m^2^, the mean gestational age was 39 ± 4.2 (weeks + days), and the mean birth weight was 3720 ± 238 g (Table 1).

To measure the distribution of C nerve fibers, histological samples were analyzed; the results showed that the IUO had a significantly larger concentration of A fibers (46 ± 4.8) than the LUS (21 ± 2.6). All these are reported in Table 2 and in Figure 2.

## 3. Discussion

Since the uterus may contain many widely dispersed pacemakers similar to the gastro-intestinal tract [16], the study concentrated on detecting the quantitative presence of adrenergic neurofibers in IUO and LUS in the pregnant uterus, assuming a physiologically active role in the stimulation of labor and birth.

For years, many studies have been conducted on uterine pacemakers, although the interaction between environment and ion channel activity remains unclear [17,18]. In the 1950s, Alvarez and Caldeyro-Barcia were the first researchers to measure intrauterine pressure in women, suggesting that the generation of high intrauterine pressures in sequential contractions is a fundamental requirement for cervical dilation in the first stage of human labor [19,20]. Reynolds, in 1948, after studies of multi-toco measurements, proposed a single heart-like uterine pacemaker with an unspecified, but fixed, location [21]. The multi-toco, intrauterine pressure catheter (IUPC), and intra-myometrial balloon studies conducted by Alvarez, Caldeyro-Barcia, et al., in 1954, suggested the possible existence of two kinds of pseudo-heart-like pacemakers, one on the fundal region and the second one near the oviducts [22]. In the late 1960s, Csapo et al. used an intrauterine sensor to propose a model of modified heart-like pacemakers with different locations, suggesting that each contraction begins with one pacemaker [23]. Wolfs and von Leeuwen conducted studies in 1979 using several intrauterine electrical sensors. They proposed a model consisting of several pacemakers in multiple locations throughout the uterus that coordinated with each other via one propagating action potential [24]. In 2016, Young and colleagues conducted investigations on mechanotransduction and electromyography. Their findings indicated the possibility of 25–30 pacemakers dispersed throughout the uterus, functioning by a process consisting of mechanotransduction, intrauterine pressure, and regional action potential propagation [25].

During labor, uterine muscle cells contract simultaneously, determining an increase in the intrauterine pressure that, combined with the fetal head pressure, leads to cervical dilatation. This might be due to the presence of sensitive pacemakers that depolarize myocytes in response to the increased intrauterine pressure, causing synchronous activity and the recruitment of additional force during contractions [26]. To activate the pacemaker and cause muscle contractions, myometrial contractions raise intrauterine pressure and wall tension. This positive feedback mechanism unites the widely spaced pacemakers [27,28].

Takeda et al. found, using a rat model, that the uterus’ electrical pacemaker activity can only initiate and propagate between tissues through mechanical-hydraulic signaling. They proposed a connection between these bioelectrical mechanosensitive pacemakers and intrauterine pressure [29]. A possible pacemaker structure has also been identified in the form of myometrial bundles projecting into the placental bed in a rat uterus; indeed, electrical mapping investigations have shown that the action potential can originate from these “myometrial-placental pacemaker zones” [30].

Myometrial tissue strips undergo contraction without the need for external stimulation because it is well known that the myometrium in various species continues to contract even after it is removed from the body. In particular, the force profiles and contraction frequencies of human myometrial strips are similar to those displayed by women going into labor. This suggests that a spontaneously active pacemaker exists in almost any small strip of the myometrium. However, it is unclear if the uterus uses different types of pacemakers during labor [31].

The role of uterine pacemakers in physiologic and pathologic labor was also suggested by Andersen et al., who suggested that abnormal uterine contraction may be the result of unusual localization of this/these uterine pacemaker/s (Figure 3) [32].

New types of uterine cells, which are similar to the interstitial cells of Cajal (ICC) of the gut, have recently been identified; they have therefore been called Cajal-like cells (about 6% of the cell types in the uterus). These Cajal-like cells are electrically passive or electrically excitable, but not spontaneously active (subsequently indicated as passive) [33,34,35,36,37]. The Cajal-like cells also exhibit an atypical inward current, which may indicate the existence of T-type calcium channels that are often found in uterine myocytes. These channels engage in activities associated with pacemakers. Determining whether interstitial cells play a role in uterine pace-making at all will take additional time [33,34,35,36,37,38,39,40]. A proposed mechanism of the Cajal-like cells states that they can link excitable smooth muscle cells, creating an electrically unstable cell grouping capable of spontaneous activity, even though both cannot oscillate on their own (Figure 1) [41].

Norepinephrine has been reported to modify the electrophysiology of the ICC in various tissues [42]. For example, in the prostate, ICC have close anatomical relationships with sympathetic adrenergic fibers. A functional relationship between ICC and nerve fibers has been found in the uterus, and their alteration has been correlated with pathological states [43,44]. In isolated ICC, adrenaline has been shown to stimulate depolarizing currents (Vh = −60 mV), which can be transmitted to electrically connected muscle cells, increasing their state of contraction [45].

These data could identify a syncytial mechanism that regulates contraction at the level of the cervix between the ICC, adrenergic neurofibers, and smooth muscle cells. The ionic basis from the norepinephrine-stimulated depolarizing current has not been identified but may be correlated with the T-type calcium current also reported in the ICC of the uterus. Of interest in this context is that it prevents this T current as mibefradil inhibits contraction, underlining the importance of the functional syncytium between adrenergic fibers, ICC, and smooth muscle cells in the myometrium (Figure 1 and Figure 3).

ICC could represent an integral part of the syncytial pacemaker that could operate at the cervical level. The presence of depolarizing T-type calcium current and calcium-activated potassium current [46] in the ICC could fuel a mechanism of oscillation of the membrane potential in physiological ranges as demonstrated by computational simulations [47]. However, the existence of this electrical activity and of other possible mechanisms that could fuel rhythmic/oscillatory activities such as those observed in hair cells remains to be experimentally verified [48,49].

The Cajal-like cells are also involved in the uterine obstetric phenomena of contractility, for example, the expulsion of menstrual debris, ascending sperm transport, embryo implantation, pregnancy, and delivery. Furthermore, López-Pingarrón et al. found a link not only between pathophysiology associated with Cajal-like cells and obstetric alterations, such as recurrent miscarriages, premature deliveries, abolition of uterine contractions, and failures of embryo implantation but also in endometriosis and leiomyoma [44]. The cervix mechano-sensitive stimulation determines the Ferguson reflex [50,51].

Immunohistochemical studies have been conducted to identify neurofibers and neurotransmitters in the pregnant uterus, particularly in the cervix and LUS. Endogenous and exogenous substances influence uterine contractions during labor; among the endogenous ones, a critical role is played by estrogen, progesterone, cortisol, oxytocin, prostaglandins, relaxin, adrenergic and cholinergic secretions, cyclic nucleotides, and calcium ions. Progesterone and estrogen are complimentary and antagonistic to each other, regulating many pathways such as the formation of gap junctions, the influx of calcium ions, the synthesis of oxytocin, adrenergic receptors, prostaglandins, and cyclic nucleotides [52]. Because of the discomfort and mental strain of labor, the body’s natural levels of adrenaline and noradrenaline rise in the mother. Additionally, the uterus has more alpha- and beta-adrenergic receptors, which change in ratio as the pregnancy progresses [53]. Thus, epinephrine and norepinephrine can stimulate, alter, and regulate myometrial activity in humans, although their physiologic and therapeutic importance is doubtful [54]. Malvasi et al. showed, in the dystocic prolonged labor, a reduction in adrenergic and noradrenergic neurofibers, determining an alteration in the vascularization of the LUS and, consequently, uterine contractility [55,56,57].

It is also known that epinephrine decreases after regional analgesia and that adrenaline and noradrenaline are tocolytic at concentrations encountered in laboring women because of a greater presence of beta-adrenergic receptors in the pregnant uterus [50]. Pain relief after combined spinal-epidural (CSE) analgesia may cause a transient impairment in the action and level of maternal catecholamine, leading to uterine hyperactivity and fetal heart rate (FHR) abnormalities [58,59].

During labor, internal and external factors such as stress, anxiety, and an undesirable atmosphere can impact the action of endogenous and exogenous hormones such as prostaglandin, oxytocin, and others. This can result in an irregular contraction rhythm and hypertonia, which can cause distress to the fetus. In particular, anxiety can alter the labor, causing prolonged childbirth, dystocia, etc., via two mechanisms described by Walter et al. [60] in their review:

β-endorphin is an endogenous opioid released during stress [61] that causes a reduction in oxytocin release, as shown in rats [62]. It inhibits the neurosecretory terminals in the neurohypophysis by binding to κ-opioid receptors [63], and reduces the pulse rate of oxytocinergic neurons of the PVN by binding to μ-opioid receptors [64];

Autonomic nervous system: Oxytocin, at term pregnancy, causes a shift in activity in the autonomic nervous system from the sympathetic to the parasympathetic nervous system, thus increasing blood flow into the uterine muscles [65] and ensuring fetal oxygen supply even during uterine contractions [66]. Stressful situations during birth increase the action of the sympathetic nervous system by activating β2 adrenoreceptors through adrenalin and noradrenalin [67,68]. In animal models, the activation of these receptors causes inhibition of uterine contractions [69].

In labor, the uterine rhythmic contractions caused by the pulsatory stress lead to a “tend-and-befriend” reaction of the mother, contrary to the usual sympathetic “fight-or-flight” reaction. Taylor et al. [70] described the biological basis of the tend-and-befriend reaction, suggesting that it is due to oxytocin and estrogen’s abilities to ensure the safety of delivery [71]. Previous studies have demonstrated the anxiolytic effect of oxytocin [72,73,74]. Moreover, oxytocin, through oxytocinergic projections connecting the hypothalamus with the hippocampus, amygdala, and prefrontal cortex, can regulate the stress response [75]. In labor, exogenous stressful situations cause the dominance of the sympathetic nervous system and, thus, a shift from the tend-and-befriend towards a fight-or-flight reaction, leading to the release of adrenaline and noradrenaline, which can slow labor progress and cause dystocia [76].

It is possible that the intermittent action of oxytocin results in a progressive increase in uterine contractions through the activation of pacemakers during physiological labor.

External or internal stimuli (e.g., fetal head malposition [77,78]) that cause stress and anxiety to the mother lead to abnormal activation of the autonomic nervous system. There are two stress mechanisms that act in opposite directions: (1) Overproduction of catecholamine corresponds to overproduction of oxytocin so that the so-called hyper-dynamic dystocia can be experienced. (2) Stress acts by increasing the endorphin levels, which causes uterine hypotonia due to a reduction in the production of oxytocin. Adrenaline leads to uterine hypertonicity through the stimulation of oxytocin, while endorphins lead to uterine hypotonicity through a reduction in oxytocin activity. In the first case, we are talking about hypertonic dystocia; in the second case, we have hypotonic dystocia (Figure 4).

In Figure 5 is our proposed mechanism of pacemaker activation: fetal head compression on the gravid cervix and IUO stimulates autonomous nerve fibers; at this point, there are two possible pathways: the physiological pathway in which there is a dominance of the parasympathetic nervous system coupled with normal oxytocin activity leading to normal stimulation and activity of cervical pacemakers and thus cervical ripening, dilatation, and eutocic delivery.

The second is the pathologic pathway in which there is abnormal activation of the sympathetic nervous system because of external or internal stress stimuli (anxiety, fetal head malposition), leading to a tocolytic effect of catecholamines and abnormal cervical pacemaker stimulation and activity, causing alterations in cervical ripening and dilatation and, thus, dystocic labor.

## 4. Materials and Methods

A total of 900 nulliparous patients were evaluated, from December 2020 to October 2023, in 3 hospitals: the Department of Gynecology and Obstetrics, “Veris Delli Ponti” Hospital, Scorrano, Lecce, Italy, and the departments of Obstetrics and Gynecology of two University-affiliated Hospitals (University of Bari and University of Perugia). Patients with a single pregnancy at term, a fetus in a cephalic presentation, and no pregnancy problems met the inclusion criteria for expectant mothers. Pregnant women with any history of gynecologic surgery as well as those with pre-eclampsia, HELLP syndrome, infections, anticoagulation therapy, ruptured membranes lasting longer than 36 h, placenta previa, and other placental pathologies were excluded from enrollment. A total of 158 women were assessed and declared eligible to participate in the study. Among them, 102 women provided informed consent and were enrolled, after transvaginal ultrasound evaluation for cervical assessment. A sagittal ultrasonographic scan to evaluate the IUO was performed on all patients (using the Aloka instrument SSD 2000 MultiView, Tokyo, Japan, and the GE Healthcare instrument, Voluson 730 Expert, Chalfont St. Giles, UK). The group was made up of patients with a singleton fetus in a cephalic presentation who were set to undergo elective CS. All patients received a prophylactic antibiotic (2 g of Cefazolin) administered intravenously before CS. The anesthesia technique was a combined spinal-epidural (CSE). The CSE technique consisted of the needle-through-needle technique in intervertebral space (L3-L4) (Espocan^®^, B. Braun, Mirandola, Italy). The spinal needle “Spinocan” 27 G was passed through the 18 G Tuohy needle. Cerebrospinal fluid was removed from the needle and a mixture of ropivacaine 0.02% with 0.25 μg/mL of Sufentanil (5 mL) was administered into the spinal space. The mixture of ropivacaine 0.07–0.15% (dilution of drugs depends on the stage, position, and station of the head) with 0.3 μg/mL of Sufentanil (10–15 mL) was administered in the epidural space. The surgical technique used was a modified Stark CS method. Uterine incision was conducted transversally on the LUS after bladder flap detachment. After fetus extraction, the placenta was delivered spontaneously and the uterus was exteriorized. The surgeons sampled two serial consecutive full-thickness sections of 5 mm depth (with the inclusion of the myometrial layer) on the LUS, using scissors, for morphological analysis. Once the IUO was identified, a small bistoury was used to obtain two samples (5 mm depth) from two random spots in the h12, h3, h6, and h9 clock-like top view of the cervix (Figure 6).

Data Analysis of Adrenergic Neurofibers. 

The specimens were immediately transferred to the laboratory in a dry-ice container, washed by immersion in cold Krebs-Ringer’s solution, and examined to detect adrenergic nerve fibers. To stain adrenergic nerve fibers, a glyoxylic acid-induced fluorescence technique was used, as described by Qayyum and Fatani [79]. Immediately before use, a staining solution called sucrose/phosphate/glyoxylic acid (SPG) was quickly prepared by adding 0.2 M sucrose and 1% glyoxylic acid to a solution of 0.236 M monobasic potassium phosphate (pH 7.4). Glyoxylic acid converts various catecholamines and indoleamines into highly fluorescent compounds and provides an improved histochemical technique for quantifying catecholaminergic neurofibers (C) [79].

Five consecutive serial sections to be checked for LUS and IUO were obtained from transverse sections of IUO and LUS samples. The 40-μm sections were obtained using cryostat microtome, placed on five separate slides, and prepared for detection of adrenergic nerve fibers. The fresh cryostatic sections were immediately dipped into the SPG solution for 5 min. After the staining was performed, the samples were washed in PBS and then mounted in Entellan (non-auto fluorescent), and examined under a Zeiss III photomicroscope (Carl Zeiss, Oberkochen, Germany) equipped with epi-Illumination and Neofluar objectives. Quantitative analysis of images (QAIs) of the density of the nerve fibers was performed using photographs of stained samples, using a Quantimet Leica 2000 image analyzer (Quan-timet 500 Leica Microsystems Imaging Solutions Ltd., Cambridge, UK) and quantified as conventional units (C.U.) that represent the area occupied by adrenergic nerve fibers compared to the total observed area. All data are presented as mean ± S.E. Statistical significance between groups was determined using Fisher’s exact test, and *p* > 0.05 was considered to be non-significant.

## 5. Conclusions

Childbirth is a very complex event that involves a series of vascular and neurotransmission functional systems associated with variable and uncertain outcomes [77,78]. One of the biggest problems that may arise during labor is dystocia, which is the cause of the biggest problems during childbirth [80].

The gravid cervix has a larger concentration of adrenergic neurofibers than the at-term LUS, according to this observational immunohistochemistry study. The presence of A neurofibers in the cervix may change how the bundles of smooth muscle cells activate, which could cause an irregular activation–deactivation pattern in the mechanosensitive pacemakers. In eutocic labor, the autonomic nervous system is dominated by the parasympathetic system, ensuring optimal blood flow to the uterus and placenta. Mechanosensitive pacemakers in the gravid cervix can function physiologically under these circumstances, resulting in legitimate uterine contractile activity and a eutocic birth. Stressful circumstances (anxiety, tension, fetal head position) cause the sympathetic nervous system to become more active, acting through these nerve fibers in the gravid cervix. They might affect the mechano-sensitive pacemakers in the cervix, which would slow down uterine contractions and cervical ripening, which could cause dystocic labor (Figure 4). When oxytocin augmentation or reduction occurs, the disruption of the oxytocin–adrenaline balanced system can result in pathological disorders such as uterine dystocia, constriction rings, and hypertonia or hypotonia with uterine inertia, which can have legal and medical consequences [80,81].

We know varied causes of labor dystocia, including incomplete uterine preparation, inadequate uterine force generation, incomplete cervical ripening, obesity, cephalopelvic disproportion, fetal malposition, infection, and maternal stress. Generally, neurological control over uterine activity is not directly considered a factor in dystocia [82]. Each of these etiologies is needed to better understand the mechanisms of labor dystocia and to develop new clinical approaches.

Our data do not exhaust the complete understanding of the mechanism of dystocia, but they open up an interesting field of investigation that has not been considered until now. Further studies and investigations are necessary to clarify this topic.

There are some limitations in this study. The sample size is small and needs to be expanded. The analysis must first include cases of physiological births in order to observe the normal distribution of adrenergic fibers in the control group. The study is also limited due to the heterogeneity of the samples analyzed. The two main clinical factors to consider are parity and the type of dystocia. From the analysis of pregnancy parity, relevant information can be extracted to understand the possible recurrence of dystocia. The type of dystocia should be stratified to obtain more useful information for the interpretation of the data.

La Rosa et al. proposed a multiscale-forward electromagnetic model of uterine contractions during pregnancy, using Maxwell’s equations and a four-compartment volume conductor geometry as a tool for helping characterize contractions using magnetomyography (MMG) and electromyography (EMG) [83]. Wang et al., in their work, describe the development and application of a human Electromyometrial imaging (EMMI) system to image and evaluate 3D uterine electrical activation patterns at high spatial and temporal resolutions during human term labor [84]. A pacemaker effect is also hypothesized for the uterus, in that pulses are generated and propagate through the entire uterus, according to a spatio-temporal pattern of electrical propagation. A multi-physics approach to represent the complex biochemo–electro–mechanical model to predict the interaction between biochemical, electrical, and mechanical fields during uterine contractions, normal functioning, and arrest, and further validation studies and implementation research will be needed to monitor the structural and functional changes in labor outcomes. Deep Learning, considered by all to be a subset of machine learning [85], and graph neural networks (GNNs) can be used to build a virtual electromyometrical model using partial derivatives as a computational approach with a space-time interconnection.

## Figures and Tables

**Figure 1 ijms-25-05630-f001:**
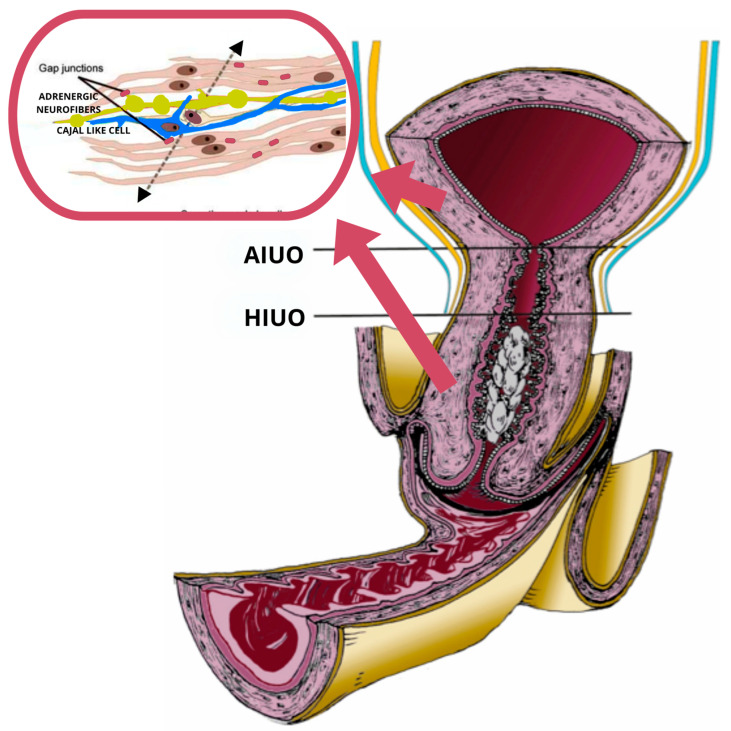
Sagittal section of the human pregnant cervix and LUS in early labor. In the red box is the smooth muscle bundle from the pregnant cervix and LUS, both regulated by adrenergic neurofibers (the yellow cell). The blue cell is the Cajal-like cell, which mediates, amplifies, and regulates bioelectrical signaling between the sensitive motor neurofibers and the smooth uterine cells through the gap junction.

**Figure 2 ijms-25-05630-f002:**
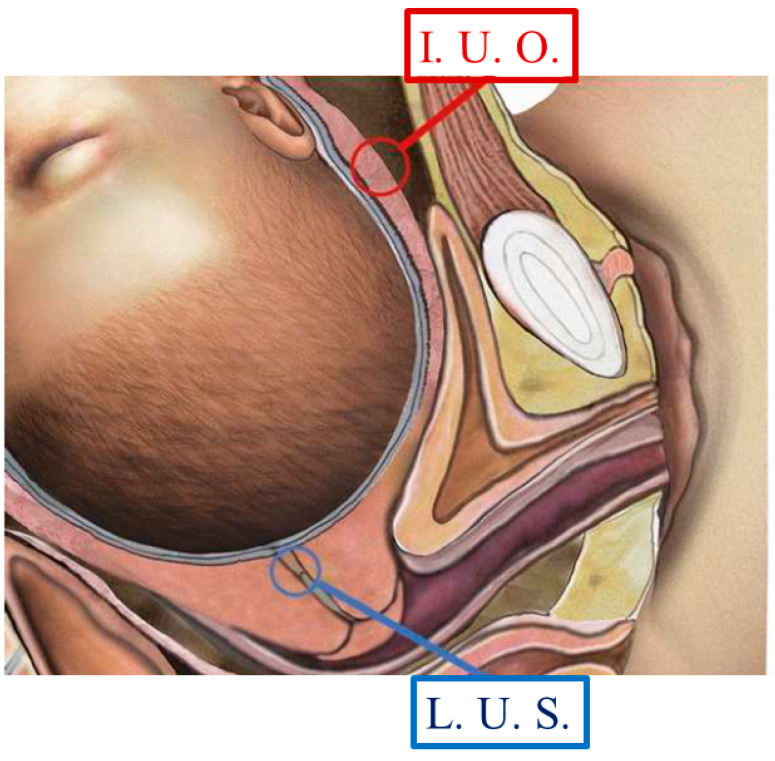
Drawing of pregnant LUS and a cervix out of labor with a fetal cephalic presentation. The red circle shows the LUS site where a biopsy was performed during the elective CS, whereas the blue circle indicates the site of the cervix of the anatomic internal uterine orifice.

**Figure 3 ijms-25-05630-f003:**
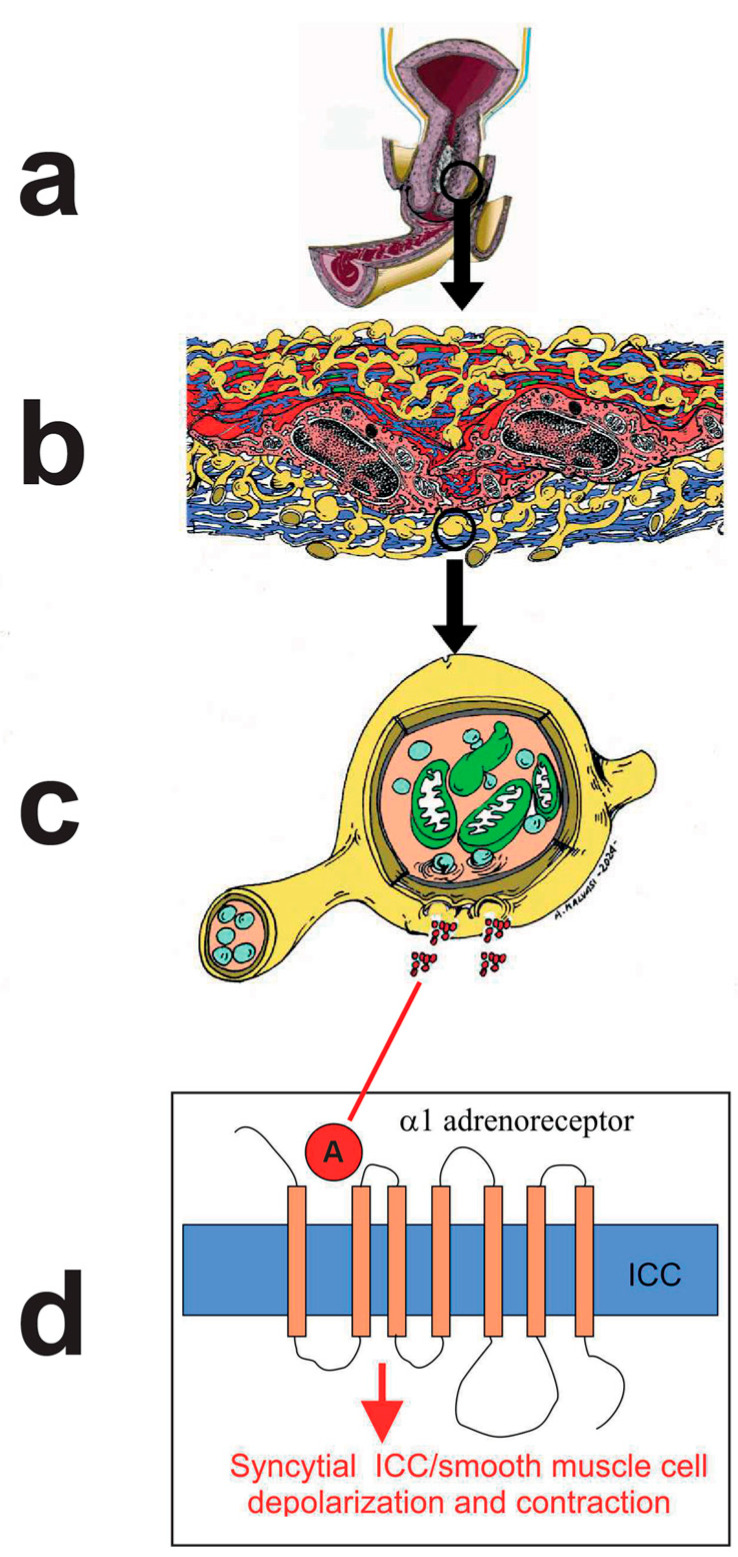
Interaction between the ICC and A neurofibers in the uterine cervix. (**a**) drawing of the cervix of an at-term uterus out of labor; (**b**) section of the muscular layer of the gravid cervix composed of smooth muscle cells (in red), Cajal-like cells/interstitial Cajal cells (ICC) (in blue), catecholamine neurofibers with varicosities (in yellow) and gap junction (in green); (**c**) an A neurofiber with a series of vesicles containing catecholamine molecules (red spheres) are released; (**d**) the release of adrenaline from the sympathetic terminals determines the activation of α1 adrenergic receptor at the level of the ICC. The activation of this receptor determines the depolarization of the membrane potential in ICC, following the activation of inward ionic current, which is transmitted through the gap junction to the syncytial smooth muscle cells. The depolarization of the smooth muscle cells determines the influx of calcium ions through the voltage-gated calcium channels and the development of the contraction process.

**Figure 4 ijms-25-05630-f004:**
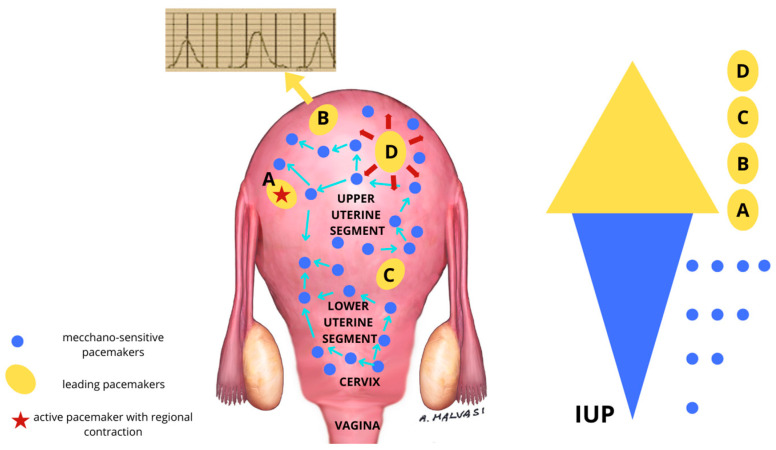
Possible localization of uterine pacemakers in a pregnant uterus in labor. The blue dots are the mechano-sensitive pacemakers innervated by sympathetic neurofibers and activated by changes in intrauterine pressure. The blue arrows represent the amplification of the regional action potential by bioelectrical propagation. The more the intrauterine pressure (IUP) increases, the more mechanosensitive pacemakers (blue dots) are recruited, as shown in the right blue and yellow graphic. The yellow ellipses are active pacemakers. The red star represents the active leading pacemaker.

**Figure 5 ijms-25-05630-f005:**
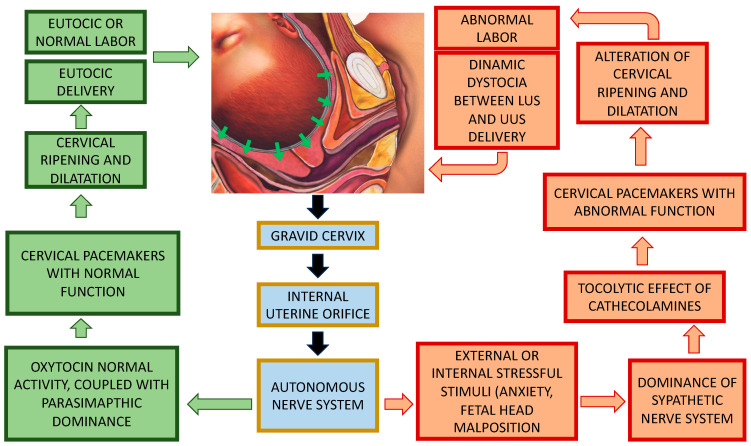
Proposed mechanism of pacemaker activation. Fetal head compression on the gravid cervix and IUO stimulates autonomous nerve fibers; at this point, there are two possible pathways: The green one represents the physiological pathway in which there is a dominance of the parasympathetic nervous system coupled with normal oxytocin activity, leading to normal stimulation and activity of cervical pacemakers and thus cervical ripening, dilatation, and eutocic delivery. The red one represents a pathologic pathway in which there is abnormal activation of the sympathetic nervous system because of external or internal stress stimuli (anxiety, fetal head malposition), leading to a tocolytic effect of catecholamines and abnormal cervical pacemaker stimulation and activity, causing alterations in cervical ripening and dilatation and, thus, dystocic labor.

**Figure 6 ijms-25-05630-f006:**
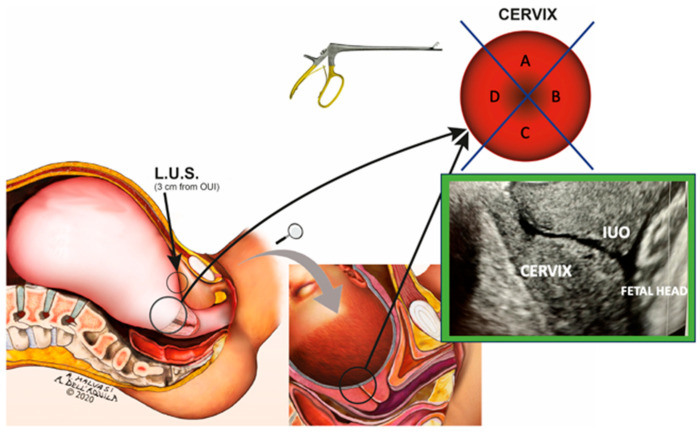
Sites of the biopsies performed during the elective cesarean section on the LUS and the gravid cervix. On the top right, a clock-like view from the top of the gravid cervix shows the cervix divided into 4 quarters A, B, C, and D. During the ECS, two samples were obtained from two random spots in these 4 quarters. The green box shows a TUS longitudinal scan of the gravid cervix; in this way, US was used to detect the IUO where the biopsies were obtained.

**Table 1 ijms-25-05630-t001:** Demographic characteristics of the patients enrolled.

Demographic Characteristics of the Patients	
N°	102
Age (years)	31.4 ± 4.4
BMI (kg/m^2^)	24.7 ± 7.3
Gestational Age (Week + Days)	39 ± 4.2
Birth Weight (g)	3720 ± 238

**Table 2 ijms-25-05630-t002:** Evaluation of catecholaminergic (C) nerve fibers in the internal uterine orifice (IUO) and lower uterine segment (LUS) of patients during elective cesarean sections (ECSs). Data are reported in C.U. (see Section 4).

Evaluation of Adrenergic Neurofibers in Gravid Uterine Cervix and LUS			
Neurofibers	I.U.O.	L.U.S.	*p*-value
Catecholamine (C)	46 ± 4.8 C.U.	21 ± 2.6 C.U.	>0.05

## Data Availability

The data presented in this study are available on request from the corresponding author, with the permission of the other coauthors.

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
