# Peer review of "Localization of Catecholaminergic Neurofibers in Pregnant Cervix as a Possible Myometrial Pacemaker"

_ijms, 2024, doi:10.3390/ijms25115630_

Round 1

Reviewer 1 Report

Comments and Suggestions for Authors

Comment 1: the discussion section is quite long, so it needs to be better articulated in order to be easier understood. I can suggest a clear paragraph organization, otherwise, please shift part of the information (for example the hystorical evolution in the knowledge on uterine pacemakers) in the introduction section.

Comment 2: line 242 can be removed or placed in another point of the discussion section?

Comment 3: line 245: the discussed substances are not peptides, please modify.

Comment 4: in the Conclusion, could you please add a sentence to clarify the impact of your findings. Moreover, a perspective paragraph should be useful to understand how your investigation will continue.

Comment 5: Please modify the caption of Figure 3 ‘the figure describes the interaction between the ICC and A neurofibers in the uterine cervix.’ With ‘Interaction between the ICC and A neurofibers in the uterine cervix’. Please use the same structure for Figure 2 and 4.

Comment 6: in the results, when you report the concentration of the A fibers, please specify the measurement unit (C.U.).

Comment 7: in Figure 1 there are overlappings. Could you please remove it?

Comment 8: in line 97, could you please add a reference?

Comment 9: In Figure 2 a ruler is required.

Comment 10: in line 205 measurement unit is required.

Comment 11: in lines  355-359, could you please measurements units?

Comment 12: line 336 remove ) and shift to line 339.

Author Response

Thank you very much for taking the time to review this manuscript. Please find the detailed responses below.

Comment 1: the discussion section is quite long, so it needs to be better articulated in order to be easier understood. I can suggest a clear paragraph organization, otherwise, please shift part of the information (for example the historical evolution in the knowledge on uterine pacemakers) in the introduction section.

Answer: we agree and perform an organization of the text in paragraph. We decided to maintain the historical part in the Discussion to have a more comprehensive perspective of the study all together.

Comment 2: line 242 can be removed or placed in another point of the discussion section?

Answer: We agree, line 242 has been shifted at the end of the previous paragraph where uterine contractility is discussed.

Comment 3: line 245: the discussed substances are not peptides, please modify.

Answer: Correct, we modified ‘peptides’ with substances.

Comment 4: in the Conclusion, could you please add a sentence to clarify the impact of your findings. Moreover, a perspective paragraph should be useful to understand how your investigation will continue.

Answer: The conclusion paragraph has been integrated with impact, limitation and perspective of this study.

Comment 5: Please modify the caption of Figure 3 ‘the figure describes the interaction between the ICC and A neurofibers in the uterine cervix.’ With ‘Interaction between the ICC and A neurofibers in the uterine cervix’. Please use the same structure for Figure 2 and 4.

Answer: the figure captions have been modified as required.

Comment 6: in the results, when you report the concentration of the A fibers, please specify the measurement unit (C.U.).

Answer: according to reviewer’s comment, the measurement unit has been specified in the title of the table.

Comment 7: in Figure 1 there are overlapping. Could you please remove it?

Answer: Ok, overlapping has been removed.

Comment 8: in line 97, could you please add a reference?

Answer: A reference has been added.

Comment 9: In Figure 2 a ruler is required.

Answer: Unfortunately, we had a problem with the computer and we lost the information necessary to produce the scale bar. For this reason, we decided to remove the example photos present in Figure 2 that do not add information in respect to the data presented in Table 2. However, we want to underline that the quantitative analysis of the images is valid, since, as described in the Methods Section, the quantification is independent from magnification since it depends on a relative comparison and it is reported in conventional units, that represent the area occupied by adrenergic fibers respect to the total observed area.

Comment 10: in line 205 measurement unit is required.

Answer: the measurement unit has been added.

Comment 11: in lines  355-359, could you please check the measurements units?

Answer: the measurement units have been checked.

Comment 12: line 336 remove ) and shift to line 339.

Answer: we modified the text, as required.

Reviewer 2 Report

Comments and Suggestions for Authors

LOCALIZATION OF ADRENALINE NEUROFIBERS IN PREGNANT - CERVIX AS A POSSIBLE MYOMETRIAL PACEMAKER.

This review  and study is  very informative to the nature of the innervation of the uterus and cervical region. Highlighting the subject of the pacemakers and the regulation  is well described and referenced. This was a pleasure to read and be informed on the topics and potential mechanisms behind pathological implications.  The figures are very well made.

                Clinicians as well as veterinary health care providers would be interested in this manuscript.

I only have minor comments for the authors :

1.      Line 51: I know it is defined in the abstract, but it would be good to define for it 1st use in the INTRODUCTION as well.  “Lower Uterine Segment (LUS)” . It is defined on line 116.

2.      Line 74 “TUS”- define.

3.      Line 254 “So, Epinephrine” no need to capitalize.

Maybe a note to the  associate editor …It would be nice if the abbreviations could be on the front page of the published manuscript.

Author Response

Thank you very much for taking the time to review this manuscript. Please find the detailed responses below and high lightened in the updated manuscript.

  1. Line 51: I know it is defined in the abstract, but it would be good to define for it 1stuse in the INTRODUCTION as well.  “Lower Uterine Segment (LUS)” . It is defined on line 116.

Answer: we defined LUS in the introduction as proposed.

  1. Line 74 “TUS”- define.

Answer: A definition for TUS has been explicated.

  1. Line 254 “So, Epinephrine” no need to capitalize.

Answer: according to reviewer’s comment, we modified the line.

Reviewer 3 Report

Comments and Suggestions for Authors

The study presents an intriguing investigation into the distribution of adrenergic (A) fibers in different regions of the pregnant uterus and its potential implications on labor dynamics. The authors aim to shed light on the role of these fibers, particularly in the context of eutocic labor and the influence of their concentration on uterine contractions and cervical ripening. Overall, the study offers valuable insights into the complex interplay between neurobiological factors and labor dynamics, with implications for understanding and managing the labor. Addressing the following areas for improvement would further enhance the clarity, depth, and relevance of the manuscript, ultimately contributing to the advancement of knowledge in obstetrics and perinatal care.

Areas for Improvement:

The study primarily relies on histological analysis to quantify the distribution of A nerve fibers. While histology provides valuable insights into tissue morphology, it would be beneficial to discuss these findings with functional assessments or physiological measurements to further elucidate the functional significance of the observed neuroanatomical differences.

The study briefly mentions stressful circumstances and their potential impact on the sympathetic nervous system activation, but a more comprehensive discussion of potential confounders and their implications on the observed findings would enhance the robustness of the conclusions. Addressing factors such as maternal anxiety levels, fetal head position, or other physiological variables could provide a more nuanced understanding of the neurobiological mechanisms at play.

While the conclusions outline several important considerations regarding the role of adrenergic innervation in labor, there is room for further clarity and precision. Providing more explicit connections between the observed neuroanatomical differences and their functional implications on labor progression would strengthen the conclusions and facilitate a clearer understanding of the study's implications.

Incorporating insights from previous studies on neuroanatomical variations, hormonal regulation, and clinical outcomes related to labor dynamics would enhance the depth of the discussion.

Including a brief discussion of potential limitations inherent in the study design, such as sample characteristics or methodological constraints, would provide transparency and context for interpreting the findings. Additionally, suggesting avenues for future research, such as exploring the interplay between adrenergic innervation and other regulatory mechanisms in labor, would contribute to advancing knowledge in the field.

Conclusion:

Overall, the study makes a valuable contribution to the understanding of neuroanatomical variations within the pregnant uterus and their potential impact on labor dynamics. Addressing the highlighted areas for improvement could further enhance the study's scientific rigor and clinical relevance.

Comments on the Quality of English Language

The manuscript should be revised by English native speaker.

Author Response

Thank you for these your fundamental observations. The article has been integrated with some information requested in the discussion, in the conclusion and a small paragraph has been added regarding the main limitations of the study.

1) The study primarily relies on histological analysis to quantify the distribution of A nerve fibers. While histology provides valuable insights into tissue morphology, it would be beneficial to discuss these findings with functional assessments or physiological measurements to further elucidate the functional significance of the observed neuroanatomical differences.

Answer: A histological and quantitative data must be confirmed from a functional point of view with all its implications. Currently our study is observational only. The results of our study support the hypothesis of abnormal activity of the sympathetic autonomic control in cases of dystocic birth. We know that during the Ferguson’s reflex, the signal is relayed by somatic spinal afferents to A2 noradrenergic neurons in the nucleus tractus solitarii (NTS) that projects to the neurons in supraoptic and paraventricular nuclei, which enhance the OXT gene transcription (https://doi.org/10.1016/j.ajog.2023.04.011). We observe that there is a quantitative difference of adrenergic fibers between the SUI and the OUI in emergency CS group, for dystocia.  This allows us to suppose a control of adrenergic fibers like a pacemaker, on the ripening of the cervix and the subsequent regulation of uterine contractile activity.

2) The study briefly mentions stressful circumstances and their potential impact on the sympathetic nervous system activation, but a more comprehensive discussion of potential confounders and their implications on the observed findings would enhance the robustness of the conclusions. Addressing factors such as maternal anxiety levels, fetal head position, or other physiological variables could provide a more nuanced understanding of the neurobiological mechanisms at play.

Answer: For over 50 years it is know that maternal stress with elevated epinephrine levels inhibits uterine contractility and lengthens labor overall [1]. Fear of childbirth [2], younger and older women (< 20 years or > 32 years at the time of first birth), and women with low social support are at higher risk for fear and stress during childbirth. A history of trauma or abuse or a bad experience during a previous labor/birth exacerbates the stress/labor association [3]. In addition, apprehension about fetal descent through the birth canal, perhaps related to maternal concern about perineal lacerations, inhibits maternal pushing efforts during the second stage. The role of maternal stress in regulating parturition physiology might contribute to women’s overall experience of prolonged labor [4].

  1. Lederman RP, Lederman E, Work BA Jr, McCann DS. The relationship of maternal anxiety, plasma catecholamines, and plasma cortisol to progress in labor. Am J Obstet Gynecol. 1978;132(5):495–500. [PubMed: 717451]
  2. Adams SS, Eberhard-Gran M, Eskild A. Fear of childbirth and duration of labor: a study of 2206 women with intended vaginal delivery. BJOG. 2012;119(10):1238–46. [PubMed: 22734617]
  3. Dencker A, Nilsson C, Begley C, Jangsten E, Mollberg M, Patel H, et al. Causes and outcomes in studies of fear of childbirth: a systematic review. Women Birth. 2019;32(2):99–111. [PubMed: 30115515]
  4. Kissler K, Jones J, McFarland AK, Luchsinger J. A qualitative meta-synthesis of women’s experiences of labor dystocia. Women Birth. 2020;33(4):e332–e8. [PubMed: 31422024]

3) While the conclusions outline several important considerations regarding the role of adrenergic innervation in labor, there is room for further clarity and precision. Providing more explicit connections between the observed neuroanatomical differences and their functional implications on labor progression would strengthen the conclusions and facilitate a clearer understanding of the study’s implications.

Answer: Every woman during labor experiences neuro-endocrine alterations such as the secretion of adrenaline, especially at the adrenal level, which influences the production of oxytocin. There are two stress mechanisms that act in opposite directions: 1) An overproduction of adrenaline corresponds to an overproduction of oxytocin such that the so-called hyper-dynamic dystocia can be experienced. 2) Stress acts by increasing the endorphin levels which determine uterine hypotonia due to a reduction in the production of oxytocin. Adrenaline leads to uterine hypertonicity through the stimulation of oxytocin, while endorphins lead to uterine hypotonicity through a reduction in oxytocin activity.  In the first case we are talking about hypertonic dystocia, in the second case instead we have a hypotonic dystocia. If we hypothesize that a number of adrenergic fibers at the OUI level can function as a pacemaker, when also external stimuli act on this point, the pacemaker does not work well.  An external stress that acts directly on the OUI can lead to hyperactivity of this pacemaker, which causes a chain activation of all the other pacemakers that are distributed throughout the uterus. In this way we encounter dynamic dystocia.

4) Incorporating insights from previous studies on neuroanatomical variations, hormonal regulation, and clinical outcomes related to labor dynamics would enhance the depth of the discussion.

Answer: The ripening of the cervix and the labor are complex process derivative of enzymatic breakdown, inflammatory and endocrine regulation. Therefore, it is apparent that cervical remodeling is a derivative of the reactions mediated by multiple factors such as hormones, prostaglandins, nitric oxide, and inflammatory cytokines. Despite a tremendous amount of research over the years, it still seems poorly understood. We still do not know if one dominant factor (or pathway) is responsible for initiating the complex biochemical changes.

Cells 2022, 11, 3690. https://doi.org/10.3390/cells11223690https://www.mdpi.com/journal/cells

Every single factor has a relevance as it cooperates with all the other elements in the dynamics of labor. Therefore, each factor is necessary as an activator, promoter or modulator of the final outcome, physiological or dystocic. For this reason, including the specific neurological information from our study, we can help to deepen and better understand such a complex problem of dystocia.

5) Including a brief discussion of potential limitations inherent in the study design, such as sample characteristics or methodological constraints, would provide transparency and context for interpreting the findings. Additionally, suggesting avenues for future research, such as exploring the interplay between adrenergic innervation and other regulatory mechanisms in labor, would contribute to advancing knowledge in the field.

Answer: Regarding the limitations of the study:

The sample is small and needs to be expanded. The analysis must first include cases of physiological births in order to observe the normal distribution of adrenergic fibers in the control group.

The study is also limited due to the heterogeneity of the sample analyzed. We are going to consider the two main clinical factors: parity and the type of dystocia.

From the analysis of pregnancy parity, relevant information can be extracted in order to understand the possible recurrence of dystocia.

The type of dystocia will be stratified to obtain useful information for the interpretation of the data. We know varied causes of labor dystocia, including incomplete uterine preparation, inadequate uterine force generation, incomplete cervical ripening, obesity, cephalopelvic disproportion, fetal malposition, infection, and maternal stress. Neurological control over uterine activity is not directly considered a factor of dystocia.

https://www.ncbi.nlm.nih.gov/pmc/articles/PMC10388369/

Each of these etiologies is needed to better understand the mechanisms of labor dystocia and to develop new clinical approaches. 

Our data do not exhaust the complete understanding of the mechanism of dystocia, but they open up an interesting field of investigation that had not been considered until now.

Further studies and investigations are necessary to clarify this topic.